# Micronutrient Deficiency as a Confounder in Ascertaining the Role of Obesity in Severe COVID-19 Infection

**DOI:** 10.3390/ijerph19031125

**Published:** 2022-01-20

**Authors:** Brian A. Chu, Vijaya Surampudi, Zhaoping Li, Christina Harris, Teresa Seeman, Keith C. Norris, Tara Vijayan

**Affiliations:** 1David Geffen School of Medicine, University of California, Los Angeles (UCLA), Los Angeles, CA 90095, USA; bachu@mednet.ucla.edu; 2Center for Human Nutrition, David Geffen School of Medicine, University of California, Los Angeles (UCLA), Los Angeles, CA 90095, USA; vsurampudi@mednet.ucla.edu; 3Department of Allergy and Immunology, Veterans Affairs Greater Los Angeles Healthcare System, Center for Human Nutrition, David Geffen School of Medicine, University of California, Los Angeles (UCLA), Los Angeles, CA 90095, USA; zli@mednet.ucla.edu; 4Department of Internal Medicine, Veterans Affairs Greater Los Angeles Healthcare System, David Geffen School of Medicine, University of California, Los Angeles (UCLA), Los Angeles, CA 90095, USA; christina.harris5@va.gov; 5Division of Geriatrics, David Geffen School of Medicine, University of California, Los Angeles (UCLA), Los Angeles, CA 90095, USA; tseeman@mednet.ucla.edu; 6Division of General Internal Medicine-Health Services Research, David Geffen School of Medicine, University of California, Los Angeles (UCLA), Los Angeles, CA 90095, USA; kcnorris@mednet.ucla.edu; 7Division of Infectious Diseases, David Geffen School of Medicine, University of California, Los Angeles (UCLA), Los Angeles, CA 90095, USA

**Keywords:** micronutrients, vitamin D, vitamin A, coronavirus, SARS-CoV-2, COVID-19, respiratory infection, obesity, food insecurity

## Abstract

Food insecurity in the United States has been exacerbated due to the socioeconomic strain of the coronavirus disease 2019 (COVID-19) pandemic. Populations experiencing poverty and, as a consequence, food insecurity in the United States are disproportionately affected by obesity, which was identified early in the pandemic as a major risk factor for increased susceptibility to COVID-19 infection and mortality. Given the focus on obesity and its role in immune dysregulation, it is also important to note the role of micronutrient deficiency, another sequalae of food insecurity. Micronutrients play an important role in the ability of the immune system to mount an appropriate response. Moreover, OBESE individuals are more likely to be micronutrient deficient. This review will explore the role of micronutrients, vitamin A, vitamin D, vitamin C, and zinc in respiratory immunity and COVID-19 and how micronutrient deficiency may be a possible confounder in obesity’s association with severe outcomes. By illuminating the role of micronutrients in COVID-19, this paper expands the discussion from food insecurity and obesity to include micronutrient deficiency and how all of these interact in respiratory illnesses such as COVID-19.

## 1. Introduction

Obesity was highlighted early in the coronavirus disease 2019 (COVID-19) pandemic as a major risk factor that predisposes patients to worse outcomes, with morbidly obese persons facing a 26% higher risk of death [1]. As a chronic inflammatory state, obesity disrupts the immune system and mechanically strains the respiratory system [2]. Populations that experience food insecurity, defined as having limited access to high-quality foods that may affect eating patterns and reduce food intake, have high rates of obesity in the United States (US) [3]. From 1999 to 2016, this population has doubled and the association between adiposity and food insecurity has increased [4]. People who are classified as being overweight or obese, defined as an increased body mass index (BMI ≥ 25 kg/m^2^ or ≥30 kg/m^2^), have a nearly two-fold increased likelihood of having COVID-19 infection or complications [5,6,7]. Richardson and colleagues reported that over 40% of patients hospitalized with COVID-19 in the New York area in March and April of 2020 were overweight (BMI ≥ 25 kg/m^2^) or obese (BMI ≥ 30 kg/m^2^) [8]. These studies and others reinforce that not only are overweight patients at a greater increased risk for COVID-19 than patients who are not overweight, but those who are obese also have a greater risk than overweight patients [9,10,11,12]. Such findings are critical for public health messaging, clinical care, and lifestyle recommendations since obesity is present in nearly half of the adult citizens in America.

Obesity is directly linked to the maldistribution of the many social determinants of health (SDoH), including but not limited to food supply, housing, economic and social relationships, transportation, employment, criminal justice, education, and health care and the programs and policies that direct them. The maldistribution of SDoH effectively determines the length and quality of life across populations [13] by having a major influence on both nutrition and SARS-CoV-2 infections [14]. Poverty is one of the most significant, yet understudied, SDoH [15]. Poverty has been defined in several ways but may be summarized as the paucity of resources required to meet basic human needs. The setting of poverty exacerbates a downstream cascade of poor health including poor nutrition, micronutrients deficiency, and obesity. The impact of poverty and malnutrition have become magnified during the COVID-19 pandemic, especially as it has become clear the economically disadvantaged are beset with greater rates of infection by SARS-CoV-2 [16]. Poverty has been overlooked recently with a large focus on prosperity associated with the rise in global wealth [17]. The inequitable distribution of economic and financial resources has conspired to widen the gap between the affluent and poor, with global extreme poverty (individual income < USD1.90/day) having increased in 2020 during the first year of the pandemic, resulting in the first increase in over 20 years reaching 9.2%, from a low of 8% in 2019 [18]. 

By global standards, the prevalence of extreme poverty in the US is very low [18]. However, among the developed nations, the US not only has one of the highest rates of poverty, it also has the worst index of social and health problems when assessed as a function of income inequality [19]. Both inequities in wealth and rates of poverty in the US vary by race and ethnicity, with most racial and ethnic minority groups having twice the prevalence of poverty, as well as obesity and COVID-19 infections and complications [20].

Income inequality in Los Angeles County (LAC) exceeds that of the United States as a whole, with 17% of the population living below the federal poverty level in 2017 [21]. It is no surprise that LAC has been one of the hardest hit regions by the COVID-19 pandemic in the US, with mortality rates in Black and Latino/a Americans nearly two and three times that of White Americans and the largest food-insecure population in the US. In LAC, as with the rest of the county, public panic resulted in consumers rushing to stockpile groceries and necessities, and many households were left without basic food and supplies [22]. In April 2020, the number of food-insecure individuals increased from one out of five to one out of every three adults in LAC [23]. COVID-19 exacerbated the conditions leading to food insecurity by interfering with access to employment, income, and education. School closures led to 600,000 students in LAC and 55 million students across the US, unable to attend school in person, [24,25,26] resulting in the 85% of children, who otherwise qualify, unable to access free school meals [27]. With limited access to free meals as well as outside activities, the pandemic further exacerbated disordered eating and obesity among youth [28,29]. Given this, much focus has shifted to understanding the role of food insecurity and obesity in COVID-19 infections and outcomes. 

However, in addition to increased adiposity, food insecurity can also lead to micronutrient deficiency [30,31,32,33]. Micronutrients are integral to the immune system’s ability to mount an appropriate response, vital to the development and expression of the body’s biological structures and processes. While micronutrient deficiency can result from food insecurity itself, it is potentiated by obesity which can lead to undernutrition in vitamins and minerals [34]. We conducted a search in Medline (via PubMed) using the keywords “COVID-19” AND “Obesity” AND “Vitamin D” OR “Vitamin A” OR “Vitamin C” OR “Zinc” OR “Micronutrients”, including underlying index terms and alternative variations of terms, to ascertain the relevant articles. The relevant studies published between 2019 and 2021 and select references citied in these articles were included. We performed a similar search in clinicaltrials.gov to gain an understanding of the number and types of registered ongoing trials in this area. We describe below the potential associations between commonly cited micronutrient deficiencies, vitamin A, vitamin D, vitamin C, and zinc and their relationship with obesity in patients with COVID-19.

## 2. Micronutrients and Severity of Respiratory Infections

### 2.1. Vitamin D

Of all the micronutrients studied, the impact of vitamin D deficiency has been evaluated by at least 92 clinical trials examining the association of vitamin D and COVID-19. Vitamin D is a fat-soluble micronutrient that comes from exposure to sunlight and diet. It undergoes metabolism in the liver to 25-OH vitamin D, the storage form of vitamin D, which is converted to calcitriol, the active form of vitamin D [35]. The ideal serum level of 25-OH D and the level at which patients are considered deficient/insufficient are controversial [36]. The Institute of Medicine (IOM) has recommended serum vitamin D concentrations should be maintained at 20–50 ng/mL, while serum 25-OH D concentrations less than 12 ng/mL are generally acknowledged as deficient as they are associated with an increased risk for bone and mineral disorders and cardiovascular and other diseases [36]. Unfortunately, the effects of vitamin D repletion on clinical outcomes such as cardiovascular disease, fractures, kidney disease progression, and mortality are not well established, but may be related in part to patient heterogeneity and study design. Recent systematic reviews suggest that vitamin D repletion is unlikely to have substantial positive impacts on mortality, cardiovascular disease, and other clinical outcomes in the general population [37], but its role in specific disease states is yet to be determined. The Endocrine Society recommends 25-OH D levels below 20 ng/mL be termed vitamin D deficiency, concentrations of 21–29 ng/mL to be termed insufficient, and normal levels should be reserved for serum 25-OH D values above 30 ng/mL [38]. Importantly, the terms 25-OH D deficiency and insufficiency are not necessarily representative of explicit disease states but a spectrum of risk for adverse outcomes related to low vitamin D levels [39]. Moreover, for a given population defined by age, race/ethnicity, and other characteristics such as co-morbidities, the prevalence and implications of vitamin D deficiency/insufficiency may vary [36]. This is highly relevant when considering the role of 25-OH D levels below 20 or ng/mL in COVID-19 positive patients who may present with other micronutrient deficiencies, oxidative stresses, and other relevant factors.

While it is best known for its role in bone metabolism, vitamin D has also been shown to be an active part of respiratory immunity. The respiratory epithelium has been found to generate a microenvironment with high levels of calcitriol due to its ability to constitutively activate vitamin D [40]. Calcitriol binding to Toll-like receptors on macrophages induces the activation of more vitamin D, the expression of more vitamin D receptors, and synthesis of downstream products [40,41]. This bolsters the efficacy of the innate immune system by increasing phagocyte activity and the oxidative burst potential of macrophages and generating antimicrobial peptides that directly kill pathogens such as bacteria [42]. Thus, vitamin D deficiency has potential biologic consequences related to increased susceptibility to respiratory infection.

One important product of the vitamin D receptor pathway is cathelicidin, a microbicidal peptide targeting intracellular pathogens such as M. tuberculosis. When serum levels of vitamin D are inadequate, usually defined as <20 ng/mL, macrophage-initiated innate immune response to M. tuberculosis is impaired, leading to an increased risk of contracting tuberculosis [43,44]. Similar findings have been reported for other upper and lower respiratory tract infections such as community-acquired pneumonia and respiratory syncytial virus (RSV), with lower levels of vitamin D being associated with an increased risk of infection [45,46,47,48,49,50,51]. When used for the treatment of influenza A H1N1, vitamin D was not found to alter viral replication or clearance, but in vitro treatment with vitamin D has been shown to decrease the expression of proinflammatory cytokines that often lead to severe complications such as pulmonary edema [52]. These findings have also been shown in RSV [53]. 

Vitamin D also has a non-traditional regulatory role in inflammation via its influence on oxidative stress pathways via nuclear factor-erythroid-2-related factor 2 (Nrf2), which regulates the expression of genes encoding antioxidant enzymes, apoptosis, inflammation, endothelial dysfunction, and cellular immunity [54,55]. Nrf2 activates the antioxidant response element (ARE), and activation of the ARE downregulates redox-sensitive and inflammatory genes, including nuclear factor-kB (NF-kB) [56]. Increasing oxidative stress leads to increased inflammation, and vice versa, and this is part of a deleterious cycle leading to the over-production of both oxidative stress and inflammation and adverse clinical sequelae [57]. Vitamin D repletion can attenuate this vicious cycle and the associated oxidative stress and inflammation by increasing Nrf2 and activating ARE. 

As a result of its immunomodulatory effects, vitamin D has emerged as a micronutrient of interest in the prevention and care of patients with COVID-19 [58]. Observational studies have found associations with low levels of circulating 25-OH D, defined as <20 ng/mL, and higher test positivity rates for COVID-19 [59]. Large metropolitan residences and air pollution have both been independently associated with lower vitamin D levels, and it remains unclear if the association between vitamin D levels and risk of COVID-19 is also mediated by other socio-structural determinants of health including limited sunlight exposure and tropospheric ozone, an air pollutant with highly reactive oxidant properties [60,61]. The complicated downstream effect of the tree canopy, which only covers about 20% of LA County [21], may also play an important role; while the tree canopy allows for individuals to spend a greater time outdoors [62], it also decreases ultraviolet light and, in theory, may also reduce the subsequent production of vitamin D from pre-vitamin D in the skin. 

As noted above, the suggestions regarding the benefits of vitamin D supplementation arise mainly from a large body of observational outcomes linking low 25-OH D levels with specific conditions, and more rigorous designs are awaited, such as Mendelian randomization studies that minimize confounding due to physical activity, outdoor exposure, diet, and other health habits and thereby advance implied causality. Clinical trials to date have not shown evidence for the benefits of the use of supplementation of vitamin D in patients with moderate-to-severe COVID-19 [63]. More well-designed, randomized controlled trials will be needed to further understand the role of vitamin D in the prevention and treatment of COVID-19 and the extent to which vitamin D might improve upon existing evidence-based interventions. It is important to note the risk of vitamin D deficiency during the pandemic was likely amplified by food insufficiency as well as the requirement to shelter at home and the wearing of face masks, both reducing exposure to sunlight and the conversion of pre-vitamin D to vitamin D [36]. It seems prudent that if vitamin D levels are assessed and determined to be low, 25-OH D supplementation (cholecalciferol or ergocalciferol) should be initiated with target serum levels maintained at 20 ng/mL or more as recommended by the IOM, or at 30 ng/mL or more as recommended by the Endocrine Society, recognizing that higher levels may be helpful in the setting of active comorbidities and likely co-existent micronutrient deficiency.

### 2.2. Vitamin A

Vitamin A, also known as retinoic acid or retinol, is another fat-soluble micronutrient, found within dairy products, liver, fish, and fortified cereals as well as various fruits and vegetables. Although vitamin A deficiency is prevalent throughout the world, it is not as common in the United States and occurs in <1% of adults. Individuals who suffer from malabsorption, such as those with histories of alcohol abuse or those with cystic fibrosis are at particularly high risk of vitamin A deficiency [64,65]. 

Vitamin A is best known for its role in vision and the prevention of night blindness. However, it also plays a key role in the immune system, helping to regulate the proliferation and differentiation of B and T cells. Within the lungs, it is important in maintaining epithelial integrity and the formation of lung alveoli [66]. 

Low retinoic acid levels have been found in other respiratory viral infections such as measles and RSV [67,68]. In the US, illness severity in measles has been associated with the degree of vitamin A deficiency [67]. As a result, the American Academy of Pediatrics and World Health Organization recommend vitamin A supplementation for patients hospitalized with measles [69]. 

While there are no studies to date that examine vitamin A levels and COVID-19, several hypotheses have emerged regarding the role of vitamin A in COVID-19. Interleukin-6 has been implicated as the one of the main cytokines contributing to disease severity in COVID-19 infection [70]. Vitamin A has been shown to attenuate cytokine release and inflammatory responses in other autoimmune conditions such as rheumatoid arthritis [71]. 

It has been hypothesized that decreased levels of vitamin A could be responsible, in part, for the immune dysfunction seen in COVID-19. COVID-19 RNA is broken down by the retinoic acid-inducible gene-I (RIG-I) pathway, which is dependent on vitamin A [72]. Due to the large size of the genome, high viral loads may overwhelm the RIG-I pathway, depleting stores of vitamin A in the body. It is possible that once this pathway no longer functions, the immune system shifts to a retinol-independent cytokine release making way for the cytokine release syndrome seen in COVID-19 [73]. 

In other systemic inflammatory reactions such as sepsis, patients have been found to have decreased levels of vitamin A [74]. More studies are needed to evaluate the role of vitamin A in the treatment of COVID-19 given its excellent safety profile, low cost, and widespread availability. While there are side effects related to hypervitaminosis A such as hepatotoxicity, pseudotumor cerebri, and rarely death, these are seen following the administration of much higher doses (>660,000 IU) than those that would be used for treatment or prevention [65,75]. 

### 2.3. Vitamin C

Vitamin C is an essential water-soluble vitamin obtained through the diet in fruits and vegetables and serves an important role in connective tissue and bone health. Deficiency is diagnosed clinically through skin and gingival signs including poor wound healing, gum bleeding, coiled hair, and perifollicular hemorrhage [76]. It can also be diagnosed with plasma levels < 0.2 mg/dl [76]. The prevalence of vitamin C deficiency varies greatly across the globe, from 74% in North India to 7.1% in the US [77]. While an inadequate diet plays a large role in deficiency, individuals who smoke or those with malabsorptive diseases are more susceptible to vitamin C deficiency [76].

As an antioxidant, vitamin C plays a role in the immune system protecting against oxidative stress due to infections and in the adrenocortical stress response enhancing cortisol release and downstream anti-inflammatory effects [78]. While a Cochrane review conducted by Hemilä et al. in 2013 found that vitamin C supplementation did not appear to decrease the incidence of the common colds in the general population, there was evidence that vitamin C intake decreased the incidence of colds in individuals under heavy short-term physical activity [79]. Given that vitamin C levels in white blood cells are decreased in the common cold [80], supplemental vitamin C during periods of increased oxidative stress such as strenuous physical activity may provide benefits. In trials with regular oral administration of vitamin C, cold duration decreased with an apparent dose dependency up to 6–8 g/day [81]. For sepsis and severe acute respiratory failure, the CITRIS-ALI trial investigated the use of intravenous vitamin C and found decreased mortality during the four-day administration of vitamin C [82,83]. Drawing from this trial, randomized control trials are currently studying the role of vitamin C in COVID-19 [78], with a pilot randomized trial in Wuhan showing decreased 28-day mortality (18% versus 50%) with high-dose vitamin C (12 g every 12 h for 7 days) [84]. 

For individuals at high risk for COVID-19 mortality and vitamin C deficiency, supplementation with vitamin C can be considered. Excessive vitamin C usually causes diarrhea and other gastrointestinal symptoms, as excess vitamin C acts as an osmotic agent [76]. There is a risk that increased long-term vitamin C intake could lead to iron overload and liver damage, but this is unlikely [76]. In severe COVID-19, vitamin C may be used as adjunctive treatment pending further data from these trials given its safety profile and low cost. It is important to note that more individuals may be at risk for vitamin C deficiency due to increased food insecurity as well as behavioral changes leading to decreased fruit and vegetable consumption [85].

### 2.4. Zinc

Zinc is an essential part of the immune system, aiding in the development and function of B and T cells and the innate immune system [86]. Zinc is also an antioxidant that helps stabilize membranes to prevent injury during the inflammatory process [87]. 

The side effects and the lack of demonstrated clinical benefit have limited its utility to date. In reviews of zinc and respiratory infection, there was indirect evidence that zinc supplementation aided the prevention of upper and lower respiratory tract infections and decreased common cold duration [88,89]. However, a recent small, randomized control trial did not demonstrate a significant difference among non-hospitalized patients receiving zinc or vitamin C supplementation [86]. Additionally, long-term supplementation has been shown to cause copper deficiency, resulting in hematologic and neurologic complications [88]. Therefore, the use of zinc is not recommended in the treatment or prevention of COVID-19 [90].

## 3. Intersection of Obesity and Micronutrient Deficiencies

As a state of chronic inflammation that leads to immune system and organ dysfunction, obesity has been documented as a major risk factor for COVID-19 morbidity and mortality. Obesity is attributed to excess caloric intake in obesogenic diets. Despite the excess calorie intake, obese individuals have a relatively high incidence of deficiency and insufficiency of several micronutrients, a phenomenon particularly prevalent in those with class III obesity, defined as having a BMI of 40 kg/m^2^ or greater [91].

Obese individuals are more likely to have lower serum 25-OH vitamin D levels, as this fat-soluble hormone partitions into body fat. Reduced 25-OH vitamin D levels have been observed in 40% to 80% of obese individuals in survey studies [92,93,94,95]. Free 25-OH vitamin D and 1,25-(OH)_2_ vitamin D have also been observed to be lower in obesity, with serum 25-OH vitamin D approximately 20% lower in obese people than in normal weight individuals [96,97].

Similarly, obese individuals may develop functional deficiencies in vitamin A with increasing adiposity. While the serum levels of vitamin A levels may be adequate, the severity of fatty liver disease was found to correlate with reductions in hepatic vitamin A levels and subsequent dysfunction in vitamin A-dependent pathways and cell signaling [98]. This “silent” vitamin A deficiency may augment susceptibility to respiratory infection, potentially leading to worse outcomes.

Moreover, vitamin C is inversely related to BMI, with obese individuals having lower levels of vitamin C [99]. While diet may play a role in lower levels of vitamin C, supplementation studies have shown that individuals of higher body weight have an attenuated response to vitamin C supplementation [100,101].

There is no clear consensus, and it remains uncertain whether the association is causal and, if so, the direction of causality. Several hypotheses have emerged describing obesity’s role in perpetuating micronutrient deficiency and insufficiency, including increased needs in relation to body size, decreased absorption, altered metabolism as a result of an underlying low-grade inflammatory processes, and sequestration within adipose tissue [102].

## 4. Discussion

With the focus on food insecurity and its sequalae during this pandemic, micronutrient deficiency should also be given as much consideration as metabolic complications. Chronic conditions linked to food insecurity, including diabetes, hypertension, chronic kidney and pulmonary diseases, high cholesterol, and even depression, are associated with increased oxidative stress and immune dysregulation, which may be worsened in the presence of obesity and micronutrient deficiency [103,104]. As noted earlier, obesity may also be associated with increased risk of COVID-19 infection and complications due to its role in mediating a pro-inflammatory state, which can lead to sub-optimal immune responses via immune system dysregulation [7]. An array of inflammatory cytokines are increased in obese tissues, such as tumor necrosis factor alpha (TNF-α), C-reactive protein (CRP), plasminogen activator inhibitor-1 (PAI-1), interleukin (IL)-6, IL-1β, CCL2, and Toll-like receptors (TLRs) of the innate immune system, which may have additional implications in responses to vaccination and viral infections [105,106]. The activation of IL-1β and IL-6 in COVID-19 has been associated with “cytokine storms”, which can have severe biological and clinical consequences [106]. 

Obesity may attenuate B- and T-cell responses, leading to both a decreased vaccination efficacy and a delay in viral resolution in infected patients. This may be due to an obesity-related hyperinflammatory state. Indeed, in obese mice, the amplification of the hyperinflammatory state by a high-fat diet led to an attenuation of vaccine-induced memory T-cell and neutralizing antibody production, as well as a more severe clinical course when exposed to the H1N1 influenza virus than non-obese mice or obese mice fed with a regular diet [107,108]. It remains unclear if obesity itself or diets rich in fat but depleted in micronutrients accounted for both outcomes. A similar clinical profile of a weakened adaptive immune response has been documented in COVID-19 patients with obesity and type 2 diabetes mellitus, supporting concerns for increased clinical risk [109,110]. This further was highlighted by Pellini et al. who found that overweight healthcare workers had significantly lower antibody titers 21 days following COVID-19 vaccination than their non-overweight peers, providing additional evidence of a potential reduction in vaccination efficacy, although the downstream clinical implications are not yet known [111]. 

Besides the activation of inflammatory cytokines and signaling factors, there are several additional pathways through which obesity may influence risk of COVID-19 infection and complications. The sequelae of being overweight and obese including metabolic, respiratory, cardiovascular, and thrombotic disorders may not only increase a patient’s risk of COVID-19 infection but also COVID-19 complications by impairing the body’s ability to cope with the initial infection [7,112]. Obesity can have a significant clinical impact due to structural changes in the body that can lead to a reduction in cardiorespiratory reserve (e.g., decreased expiratory reserve volume and functional residual capacity), thereby decreasing cardiorespiratory fitness and increasing susceptibility to immune-driven vascular and thrombotic effects [112]. 

Moreover, lockdown and the promotion of isolation during the pandemic have resulted in changes in dietary patterns, with people snacking more and choosing foods with lower nutritional value [85]. In overweight and obese individuals, disruptive eating behaviors have increased [113]. Coupled with the significant increase in people experiencing food insecurity, these dietary behaviors have led to an increased consumption of nutritionally deficient diets over the past year. 

Los Angeles County (LAC) has over 10 million residents (more than 42 states) and is often considered an exemplar of nation health trends and profiles. In LAC, more than 1.9 million people are newly food insecure due to the consequences of COVID-19 [22]. It is estimated that in 2020, LAC had 6.2 million people living in food-insecure households, the highest number within the US [22]. South, Central, and East Los Angeles are also known food deserts, where residents’ opportunities to choose and sustain healthier diets are limited by access to healthier food services [114,115]. In South Los Angeles, around 94% of food retail stores are corner and convenience stores [115]. Minority populations are disproportionately affected, with over 76% of LAC’s Latino/a population living in these areas [116]. Additionally, they make up over two-thirds of the county’s food-insecure households [117]. Geographically, these areas have also been found to be clusters of high positivity rates for COVID-19 [118]. The effects of structural racism are apparent in the higher test positivity in those of Latino/a race/ethnicity, with the disproportionate contribution of the Latino/a population to the essential job sector from healthcare to grocery stores, as well as household density, poverty, and lower levels of educational attainment [118]. These are the same risk factors for food insecurity. In a survey conducted by the LA Department of Public Health, over 70% of food insecure adults did not have a bachelor’s degree and over half of food insecure adults were unemployed or not in the labor force [117]. The risk factors that predispose individuals to infectious diseases and food insecurity overlap, intertwined within the lived experiences of those affected. As COVID-19 has shown, food insecurity and poverty set the stage for illness, which further perpetuates barriers of social mobility, unemployment, and deferring or forgoing education due to financial hardship.

Currently, the COVID-19 pandemic has highlighted the intersection and potential devastation of malnutrition including overweight, obesity, and micronutrient deficiency and viral respiratory disorders even in the US, a well-resourced country. Our emerging understanding of cardiometabolic disease in general and overweight/obesity being compounded by micronutrient deficiency as major risk factors for COVID-19 infection, hospitalization, and death has led to a better understanding of the pathophysiology of COVID-19 infections and important public health messaging to address COVID-19 across the nation. Unfortunately, the burden of overweight/obesity and micronutrient deficiency falls disproportionately on low-income populations and predominately racial and ethnic minorities, and the increased rates of COVID 19 infections, hospitalizations, and deaths among marginalized communities mirror the epidemiology of the increased rates of COVID-19 infections, hospitalizations, and deaths noted in patients who are overweight/obese. Educational efforts addressing poor dietary intake, overweight and obesity, and now also vaccination to reduce the risk of COVID-19 infections and its complications have met resistance related to intense counter-messaging that influences behavioral choices and that are not in an individual’s best interest. Ongoing community-level efforts addressing both overweight and obesity and COVID-19 need to be pursued and reinforced while ensuring that respectful yet accurate messaging reaches the communities in greatest need [20,105,106]. 

The limitations of this study include the paucity of long-term randomized trials regarding micronutrients in patients with COVID-19 and obesity. In addition, the role of micronutrient deficiency in acute infections such as COVID-19 is difficult to ascribe to a given vitamin or nutrient due to their interdependence, limiting the assignment of risk and treatment recommendation based on RCT. However, given their high safety profile and low cost and intimate involvement in the inflammatory and immune processes, the recommendation for their supplementation to normal levels seems prudent. The emerging evidence points to the overweight/obesity proinflammatory state as a major factor leading to the attenuation of the normal activation of the immune system in COVID-19 patients, resulting in worse outcomes. As efforts to increase vaccine uptake continue, it is important to reinforce the message that overweight/obesity and resultant micronutrient deficiencies are important risk factors, and research studies need to continue to monitor the COVID-19 vaccine immune response in high-risk populations including those with overweight/obesity and micronutrient deficiency, and to recognize that this clinical profile is even more common in low-income and racial and ethnic minority groups. This monitoring should include checking for early clinical signs of COVID-19 in overweight/obese persons so that early intervention can be promptly managed [7]. Close attention to micronutrient deficiency is also warranted. From maintaining the respiratory epithelial barrier to ensuring proper signaling in the immune response, micronutrient deficiency may be associated with worse outcomes in COVID-19 and other respiratory infections. Such undervalued consequences of food insecurity and, more specifically, the intake of poor-quality food and micronutrient deficiency in the context of obesity, warrant further exploration.

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
