# Peer review of "Micronutrient Deficiency as a Confounder in Ascertaining the Role of Obesity in Severe COVID-19 Infection"

_ijerph, 2022, doi:10.3390/ijerph19031125_

Round 1

Reviewer 1 Report

Chu, et al. reviewed published data on the impact of obesity and some of the micronutrients on the severity of COVID-19 in the context of the socio-economic status of the affected populations, trying to draw connections among the three. The connections among obesity, micronutrients and COVID-19 have been extensively reviewed during the past 18 months. A quick pubmed search reveals 39 review articles on the subject. The key question is what new ideas this manuscript is able to provide that has not been reported before and that provide significant insights to a well discussed hot topic. The reviewer find that the connections between obesity and some of the micronutrient deficiencies might be the unique angle of this manuscript. The next question is what the new findings are regarding the connections. The authors cited about 7 studies to make their points.

Overall, the manuscript is well written. However, the authors failed to do a thorough proof-reading and, as a result, the manuscript is littered with typos, textual errors, and perhaps grammatical errors due to carelessness. Such errors need to be fixed before the manuscript is deemed as publishable. Also, this reviewer has not done plagiarism check in the hope that the editor(s) will do it with their powerful tools available.

Author Response

Reviewer 1:

Chu, et al. reviewed published data on the impact of obesity and some of the micronutrients on the severity of COVID-19 in the context of the socio-economic status of the affected populations, trying to draw connections among the three. The connections among obesity, micronutrients and COVID-19 have been extensively reviewed during the past 18 months. A quick pubmed search reveals 39 review articles on the subject. The key question is what new ideas this manuscript is able to provide that has not been reported before and that provide significant insights to a well discussed hot topic. The reviewer find that the connections between obesity and some of the micronutrient deficiencies might be the unique angle of this manuscript. The next question is what the new findings are regarding the connections. The authors cited about 7 studies to make their points.

Overall, the manuscript is well written. However, the authors failed to do a thorough proof-reading and, as a result, the manuscript is littered with typos, textual errors, and perhaps grammatical errors due to carelessness. Such errors need to be fixed before the manuscript is deemed as publishable. Also, this reviewer has not done plagiarism check in the hope that the editor(s) will do it with their powerful tools available.

Thank you and we apologize for the lack of attention to those details and have now addressed them.

Reviewer 2 Report

This review summarizes the current state of the topic under study.
Furthermore, the review provide a summary of what the authors believe are the best and most relevant prior publications. The review in question is very complete in all its parts and I think it gives an excellent starting point to further explore the topic. I believe it can be published in this form.

more specific comments

ABSTRACT: It would be appropriate to express the scientific contribution of the review article to the scientific literature.
INTRODUCTION: It would be advisable to add to the introductory part a specific subsection for the literature review that has been carried out.
DISCUSSION: It is advisable to underline the limitations that this review may have, starting from the concept that the review work is the basis for the scientific debate on the topic

Author Response

Reviewer 2:

This review summarizes the current state of the topic under study.
Furthermore, the review provide a summary of what the authors believe are the best and most relevant prior publications. The review in question is very complete in all its parts and I think it gives an excellent starting point to further explore the topic. I believe it can be published in this form.

Thank you for the positive critique.

more specific comments

ABSTRACT: It would be appropriate to express the scientific contribution of the review article to the scientific literature.
INTRODUCTION: It would be advisable to add to the introductory part a specific subsection for the literature review that has been carried out.
DISCUSSION: It is advisable to underline the limitations that this review may have, starting from the concept that the review work is the basis for the scientific debate on the topic

Thank you and we have incorporated these suggestions. 

Reviewer 3 Report

General comments: Thank you for the opportunity to review this manuscript title “Micronutrient Deficiency as a Confounder in Ascertaining Role of Obesity in Severe COVID-19 Infection”. First of all, I congratulate you for your work. I find this study very interesting and it focus on a really important topic, relevant to public and global health. The review is clear, comprehensive and of relevance to the field.

However, please see my comments below:

Introduction:

  • The authors made an accurate introduction explaining how social determinants of health are related to the higher prevalence of obesity and overweight and their great influence on both nutrition and SARS CoV-2 infections. However, it seems necessary to highlight the aim of this study or the reason for doing it.

Micronutrients and severity of respiratory Infections

  • In this section the authors made a valuable research about the impact and the association of some micronutrients and COVID-19. However, I am unsure why the authors decide to include only these vitamins. For example, vitamin C is related to respiratory diseases and micronutrient deficiencies, but the authors just mention this vitamin when they write about zinc (page 5 line 239). I suggest the authors include this vitamin in this section and I also encourage them to include other micronutrients related to respiratory diseases, malnutrition and obesity/overweight.

References:

  • Page 8 line 390: After the first author’s name (Cariou B.) there is an additional number 1, please remove it.

Author Response

Reviewer 3:

Thank you for the opportunity to review this manuscript title “Micronutrient Deficiency as a Confounder in Ascertaining Role of Obesity in Severe COVID-19 Infection”. First of all, I congratulate you for your work. I find this study very interesting and it focus on a really important topic, relevant to public and global health. The review is clear, comprehensive and of relevance to the field.

Thank you for the positive critique.

However, please see my comments below:

Introduction:

  • The authors made an accurate introduction explaining how social determinants of health are related to the higher prevalence of obesity and overweight and their great influence on both nutrition and SARS CoV-2 infections. However, it seems necessary to highlight the aim of this study or the reason for doing it.

Micronutrients and severity of respiratory Infections

  • In this section the authors made a valuable research about the impact and the association of some micronutrients and COVID-19. However, I am unsure why the authors decide to include only these vitamins. For example, vitamin C is related to respiratory diseases and micronutrient deficiencies, but the authors just mention this vitamin when they write about zinc (page 5 line 239). I suggest the authors include this vitamin in this section and I also encourage them to include other micronutrients related to respiratory diseases, malnutrition and obesity/overweight.

We agree. Vitamin C is an important micronutrient in respiratory illness and obesity. We have added in a section on vitamin C. 

References:

  • Page 8 line 390: After the first author’s name (Cariou B.) there is an additional number 1, please remove it.

Thank you for catching this. This has been removed.